# Assessing insect responses to climate change: What are we testing for? Where should we be heading?

Nigel R. Andrew[1,2], Sarah J. Hill[2], Matthew Binns[1,2], Md Habibullah Bahar[1,5], Emma V. Ridley[3], Myung-Pyo Jung[1,4], Chris Fyfe[2], Michelle Yates[1,2] and Mohammad Khusro[1]

[1] Centre for Behavioural and Physiological Ecology, Zoology, University of New England, Armidale, Australia
[2] School of Environmental and Rural Sciences, University of New England, Armidale, Australia
[3] Department of Biology, University of York, York, UK
[4] Department of Agricultural Biology, National Academy of Agricultural Science, Suwon, South Korea
[5] Saskatoon Research Centre, Agriculture and Agri-Food Canada, Saskatoon, Canada

Corresponding author
Nigel R. Andrew,
nigel.andrew@une.edu.au

## ABSTRACT

To understand how researchers are tackling globally important issues, it is crucial to identify whether current research is comprehensive enough to make substantive predictions about general responses. We examined how research on climate change affecting insects is being assessed, what factors are being tested and the localities of studies, from 1703 papers published between 1985 and August 2012. Most published research (64%) is generated from Europe and North America and being dedicated to core data analysis, with 29% of the studies analysed dedicated to Lepidoptera and 22% Diptera: which are well above their contribution to the currently identified insect species richness (estimated at 13% and 17% respectively). Research publications on Coleoptera fall well short of their proportional contribution (19% of publications but 39% of insect species identified), and to a lesser extent so do Hemiptera, and Hymenoptera. Species specific responses to changes in temperature by assessing distribution/range shifts or changes in abundance were the most commonly used methods of assessing the impact of climate change on insects. Research on insects and climate change to date is dominated by manuscripts assessing butterflies in Europe, insects of economic and/or environmental concern in forestry, agriculture, and model organisms. The research on understanding how insects will respond to a rapidly changing climate is still in its infancy, but the current trends of publications give a good basis for how we are attempting to assess insect responses. In particular, there is a crucial need for broader studies of ecological, behavioural, physiological and life history responses to be addressed across a greater range of geographic locations, particularly Asia, Africa and Australasia, and in areas of high human population growth and habitat modification. It is still too early in our understanding of taxa responses to climate change to know if charismatic taxa, such as butterflies, or disease vectors, including Diptera, can be used as keystone taxa to generalise other insect responses to climate change. This is critical as the basic biology of most species is still poorly known, and dominant, well studied taxa may show variable responses to climate change across their distribution due to regional biotic and abiotic

influences. Indeed identifying if insect responses to climate change can be generalised using phylogeny, functional traits, or functional groups, or will populations and species exhibit idiosyncratic responses, should be a key priority for future research.

## INTRODUCTION

Attempts to assess and predict the biological impacts of climate change, are two of the most important and intensive research endeavours of our time (*Parmesan & Yohe, 2003*; *Root et al., 2003*; *IPCC, 2007*; *Loarie et al., 2009*). The knowledge base for the information relating to biological impacts of climate change is highly integrative, covering broad disciplines of behaviour, biogeography, ecology, evolution, genetics, genomics, phenology and physiology (*Hoffmann, Sørensen & Loeschcke, 2003*; *Andrew & Hughes, 2004*; *Andrew & Hughes, 2005*; *Parmesan, 2007*; *Clusella-Trullas et al., 2008*; *Williams et al., 2008*; *Kellermann et al., 2009*; *Chown et al., 2010*; *Andrew, Hart & Terblanche, 2011*; *Burthe et al., 2011*). In spite of the large amounts of data that are being accumulated addressing biotic responses of organisms to climate change, we are still a long way from understanding whether there are generalities across space, taxa, and time in terms of responses and adaptability to rapid change, or whether most organisms will respond idiosyncratically, and the implications of this on species interactions both within and among trophic levels. In addition, the lack of long-term records ($>$50 years) means that much of the research today is being conducted without a strong baseline to assess species and community responses to climate change, with the exception of a few charismatic species in some regions (*Woodward, 1987*; *Keatley et al., 2002*; *Peñuelas, Filella & Comas, 2002*; *Brereton et al., 2011*; *Moore, Thompson & Hawkins, 2011*; *Rafferty & Ives, 2011*; *Wilson & Maclean, 2011*; *Andrew, 2013*).

Insects are the most dominant groups of organisms on the planet in terms of species richness, abundance and biomass (*Gullan & Cranston, 2010*). The crucial roles that they play in all terrestrial and many aquatic ecosystems, are well understood and are critical for ecological function and food web structure (*Price et al., 2011*). However, our understanding of individual species is extremely limited. Most species of insects are yet to be identified (*Yeates, Harvey & Austin, 2003*; *Cranston, 2010*), and many have had little, if any, biological information collected on them, yet many species are potentially highly vulnerable to the impacts of climate change and extinction (*Dunn, 2005*). Nevertheless, there has been a substantive amount of research dedicated to assessing potential insect responses to human-induced climate change. One of the major challenges we face is to determine the impacts of climate change on insects, broadly, and gather enough information to develop mechanistic models on single species (*Angilletta, 2009*; *Kearney et al., 2009*) and species interactions (e.g. game theory *Mitchell & Angilletta, 2009*) across a range of species to assess multi-trophic community responses to climate change.

Assessing how the research community is attempting to measure climate change impacts can be done by identifying research agendas within publications and identifying trends and gaps across published literature. Here we assessed published research on climate change and insects to assess the type of research being carried out, its geographic location and identify the Orders, variables and factors being tested. Our intention was not to review the findings of the studies themselves, as has been done by successive high profile publications and meta-analyses (*Walther et al., 2002*; *Hughes, 2003*; *Parmesan & Yohe, 2003*; *Root et al., 2003*; *Parmesan, 2006*; *Williams et al., 2008*; *Loarie et al., 2009*), but to identify questions posed and assumptions made in the published literature on insect responses to climate change.

## MATERIALS AND METHODS

A literature search was conducted on all available published literature up to the 2nd August 2012 in Scopus (http://www.scopus.com) for literature with the keywords: "clim⋆ change and insect", "clim⋆ change and [each of the insect order names and generally used common names]". Citations and abstracts were downloaded for 4704 studies. Studies that had these keywords ranged in years from 1924 to online early publications (August 2012). Each abstract was then examined to determine if it was appropriate for further assessment, and a PDF reprint was then downloaded. Scopus also adds keywords in addition to the authors' keywords, with many articles not actually referring to climate change, in any part of their study. If climate change (or equivalent term) was not referred to in the study (abstract or main body) it was not included in the assessment. The number of publications was then culled down to 1703 ranging in years from 1985 to online early (2nd August 2012). If studies were deemed appropriate, a range of information was extracted as shown in Table 1. Number of species in each order that have been formally identified as of 2010 was generated from *Zborowski & Storey (2010)*. The papers assessed and data extracted was uploaded onto FigShare http://dx.doi.org/10.6084/m9.figshare.105599.

## RESULTS AND DISCUSSION

Since the mid-1990's there has been a rise in climate change and insect related publications relative to previous years (Fig. 1), with a substantive increase in 2006. By early August 2012, 127 publications had already been published in that year, so it is expected that the increasing trend will continue for the remainder of 2012.

The highest percentage of studies focused on Lepidoptera, Diptera, and Coleoptera (Fig. 2a). When the number of studies were compared to the identified species richness within each Order, Lepidoptera, Diptera, Orthoptera, Collembola, and Odonata have a proportionally higher percentage of papers assessing their responses to climate change relative to number of species identified, whilst Coleoptera, Hymenoptera, and Hemiptera have a proportionally lower percentage of papers assessing their responses to climate change relative to number of species identified. Lepidoptera were most commonly studied Order in "data" studies, "modeling" studies, "anecdotal" and "desktop" studies (Fig. 2b). For "review" studies, "multiple" Orders were the most dominant.

**Table 1** Categories given to each study for data type, region, the main climatic drivers that authors identified, the type of information that authors collected and presented in their results, and the habitat in which the study was carried out.

| Data type | Region | Climatic drivers | Information | Habitat |
|---|---|---|---|---|
| Data only | Africa | Temperature (Temp) | Abundance | Native |
| Data and modelling | Antarctic | Moisture | Distribution/range shift | Agricultural |
| Desktop | Arctic | Temp and Moist | Interactions | Native/Agricultural |
| Modelling | Asia | Evolution | Assemblage composition | Forestry |
| Review | Australia/Oceania | Carbon dioxide ($CO_2$) | Phenology | Human/Domestic |
| No Theme | Europe | Temp and $CO_2$ | Development time | Animal |
| | Global | Variety | Survival | Non-specific |
| | Middle East | Non specific | Physiology | |
| | New World | Fire | Non-specific | |
| | Non-specific | $CO_2$ and Ozone | Genetics/Genomics | |
| | North America | UVB | Behaviour | |
| | South America | Others | Morphology | |
| | Tropics | | Body weight | |
| | Variety | | Other life history traits | |

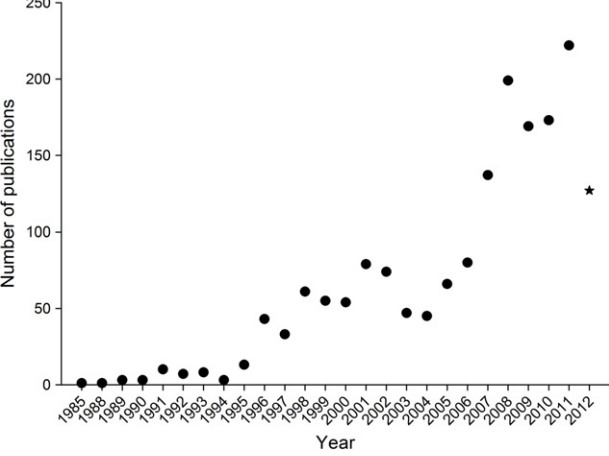

**Figure 1** Number of publications assessing the impact of climate change on insects from 1985 to 2012. A star is shown for 2012 as it only includes papers up to August 2nd.

Climate change research assessing insects was most dominant in Europe and North America (Fig. 3). Lepidoptera were by far the most dominant Order studied in Europe (Fig. 3a), and most dominant in North America (Fig. 3b), as well as in Asia (but with a similar proportion of studies published on Diptera; Fig. 3d). In Australia/Oceania, Africa, and South America, Diptera were the most highly studied Order (Figs. 3e–3g). When studies were conducted across a few regions ('variety'), generally multiple Orders were assessed (Fig. 3h). When no geographic region was identified, the specific Orders assessed were also not clearly identified (Fig. 3c).

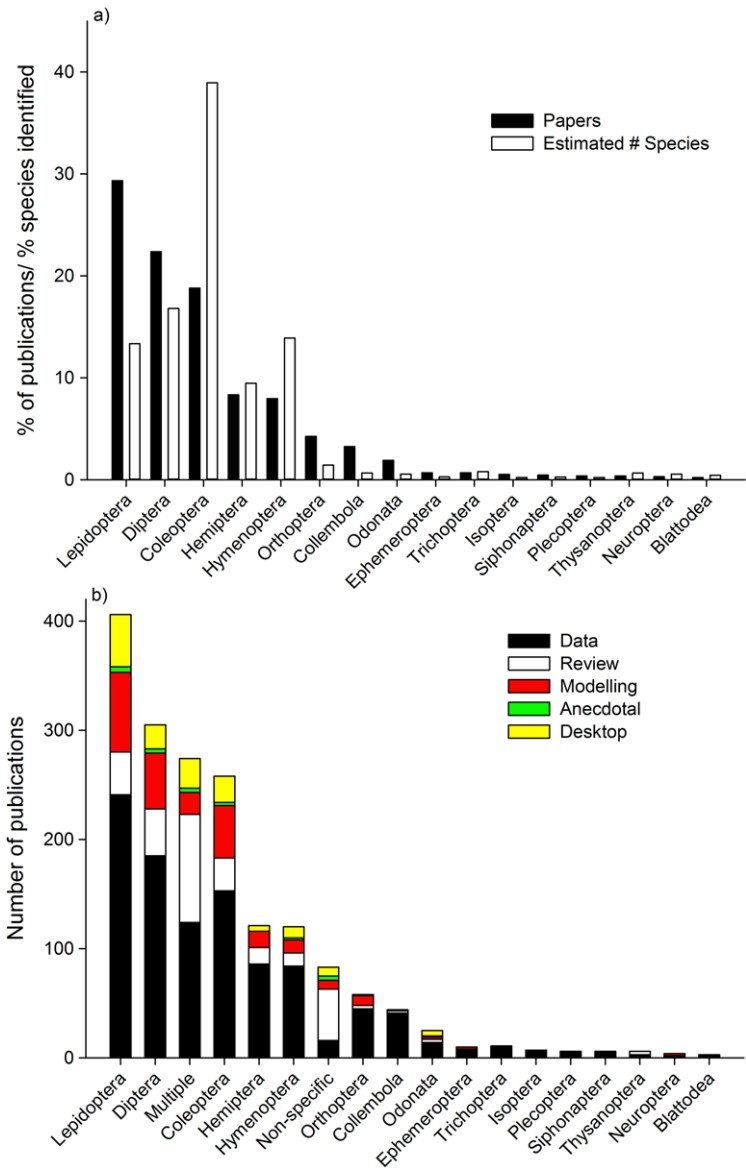

**Figure 2** (a) Proportion of published papers ($n = 1703$) and estimated number of species (Zboroski 2010; $n = 898\,730$ species) within the top 18 orders studied. (b) Number of published papers in each of the top 18 order studies, and publication type. Data type based on Table 1.

Temperature was the most studied climate change factor (40% of publications; Fig. 4). Surprisingly, the second most dominant climate change factor in studies was 'non specific' (27%), indicating that many studies mentioned climate change but did not identify a specific aspect which elicited a biotic response. The third most common factor assessed was moisture (14%), and the fourth was those assessing more than two climate change drivers (variety; 10%), including combinations of temperature, moisture, carbon dioxide, ozone, and UVB among others. Evolutionary changes, other abiotic predictors (such as UVB, ozone) and host plant changes (ie indirect changes to insects) were tested in a relatively

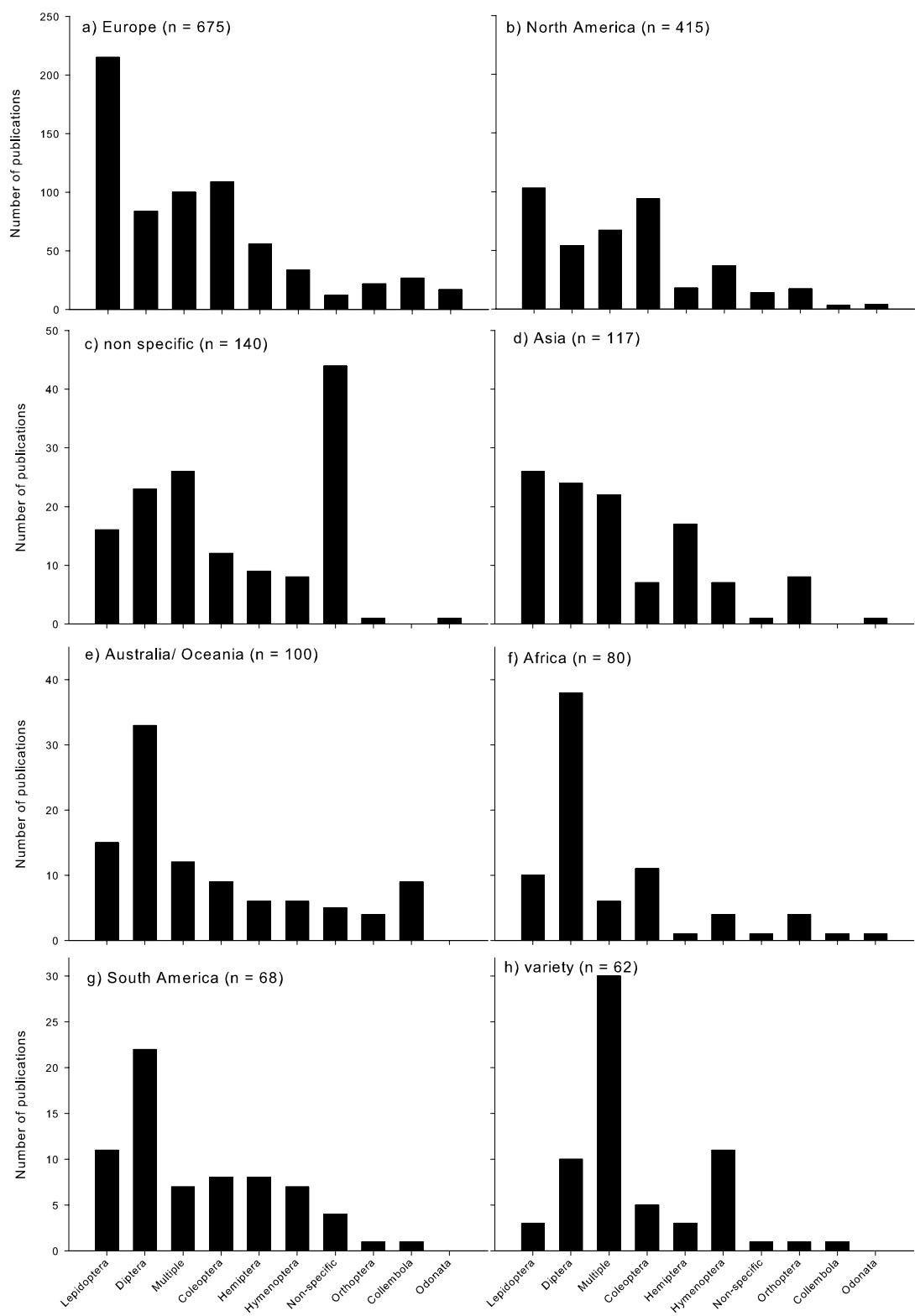

**Figure 3** Number of published studies assessing the impacts of climate change on the numerically top insect Orders (based on number of publications) from different global regions. Regions based on Table 1.

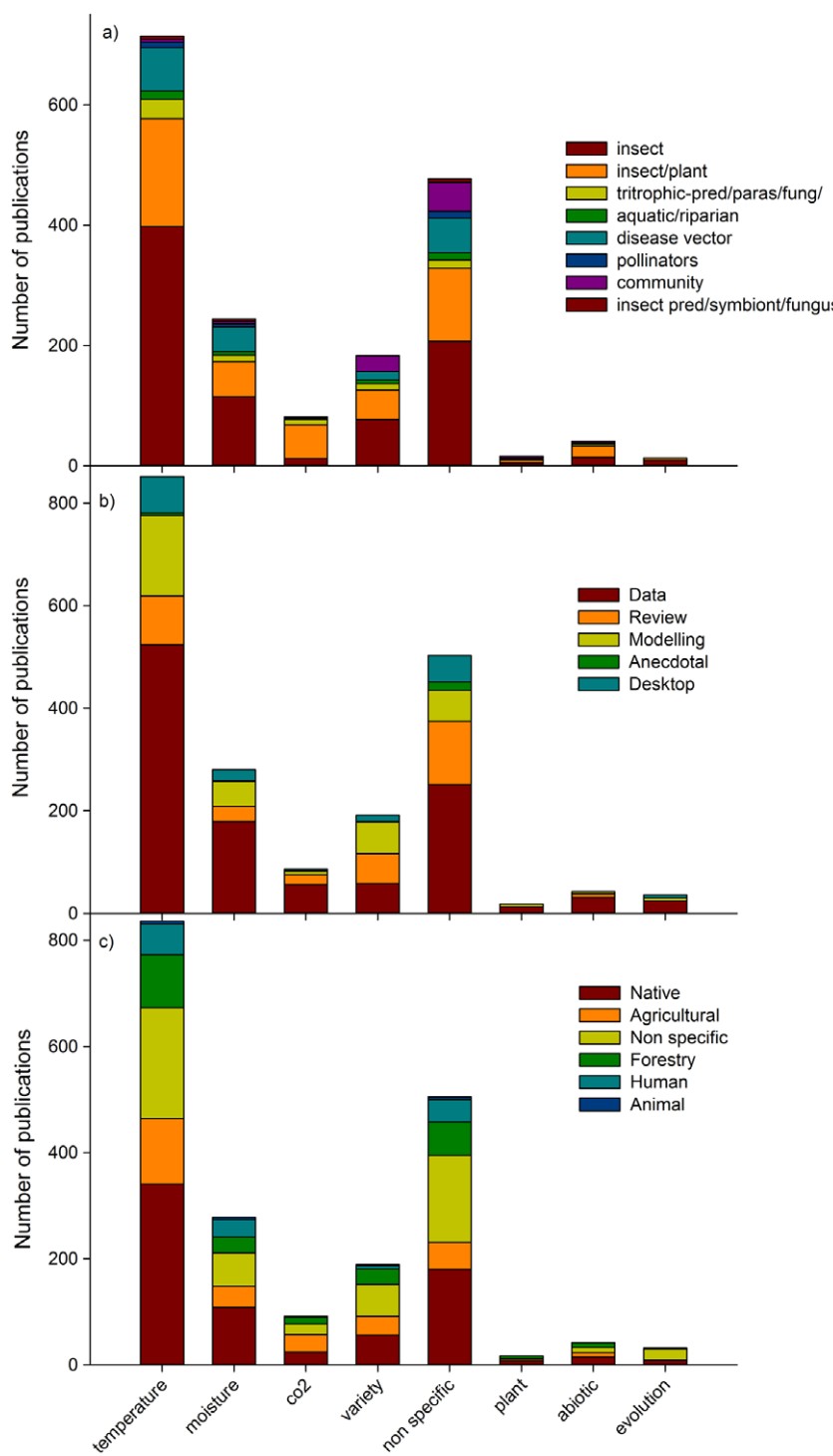

**Figure 4** Number of publications addressing different climate change factors by (a) assemblage type; (b) publication type and (c) habitat type. Groupings based on Table 1.

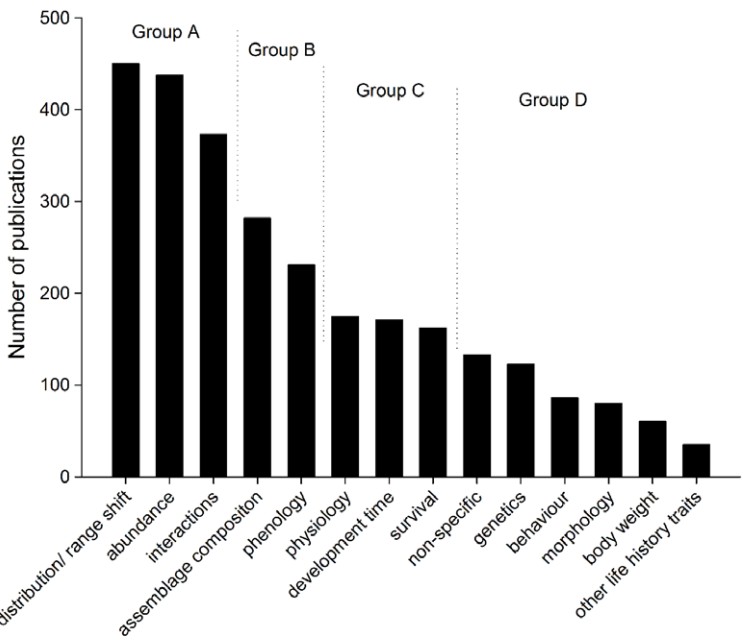

**Figure 5** How insect responses to climate change have been recorded in publications between 1985 and 2012. Four groups allocated (A–D) based on number of publications in each response group (Table 1).

small number (4% in total) of studies. Individual insect species publications were the most common (47% of publications), followed by insect/plant publications (28%) (Fig. 4a). In terms of publication type, data papers (56% of publications), dominated all predictor categories, with reviews second most common in 'non-specific' predictor papers, whereas modelling papers (17% of total publications) were more prevalent in papers dealing with temperature, moisture and a variety of predictors (Fig. 4b). Most studies conducted in native habitat assessed temperature changes (17% of total publications; Fig. 4c).

The variables that were used to measure insect responses to climate change in the published literature could be broken into four groups (Fig. 5). Most publications on insect responses to climate change (more than 390 publications in each category, Group A) assessed changes in distribution or range shift, changes in abundance and interactions (such as herbivory, predation, and parasitism). The second group of publications (Group B, above 250 and less than 310 publications each) assessed assemblage composition changes and phenology, and Group C (above 175 and less than 200 publications each) assessed insect physiology, development time, and survival. Group D (less than 150 publications each) included papers where climate change was mentioned, but where no direct assessment was carried out (non-specific), as well as assessments of genetics, behaviour, morphology, body weight changes, and other life history traits.

Most of the studies were conducted in native habitats (38% of publications: Fig. 6a) or where habitat was unspecified (28%: Fig. 6b). Forestry and agricultural habitats had substantially less studies (15% and 12% respectively; Figs. 6c and 6d) followed by studies assessing human interactions with insects and climate change (Fig. 6e), and livestock

**Peer**J

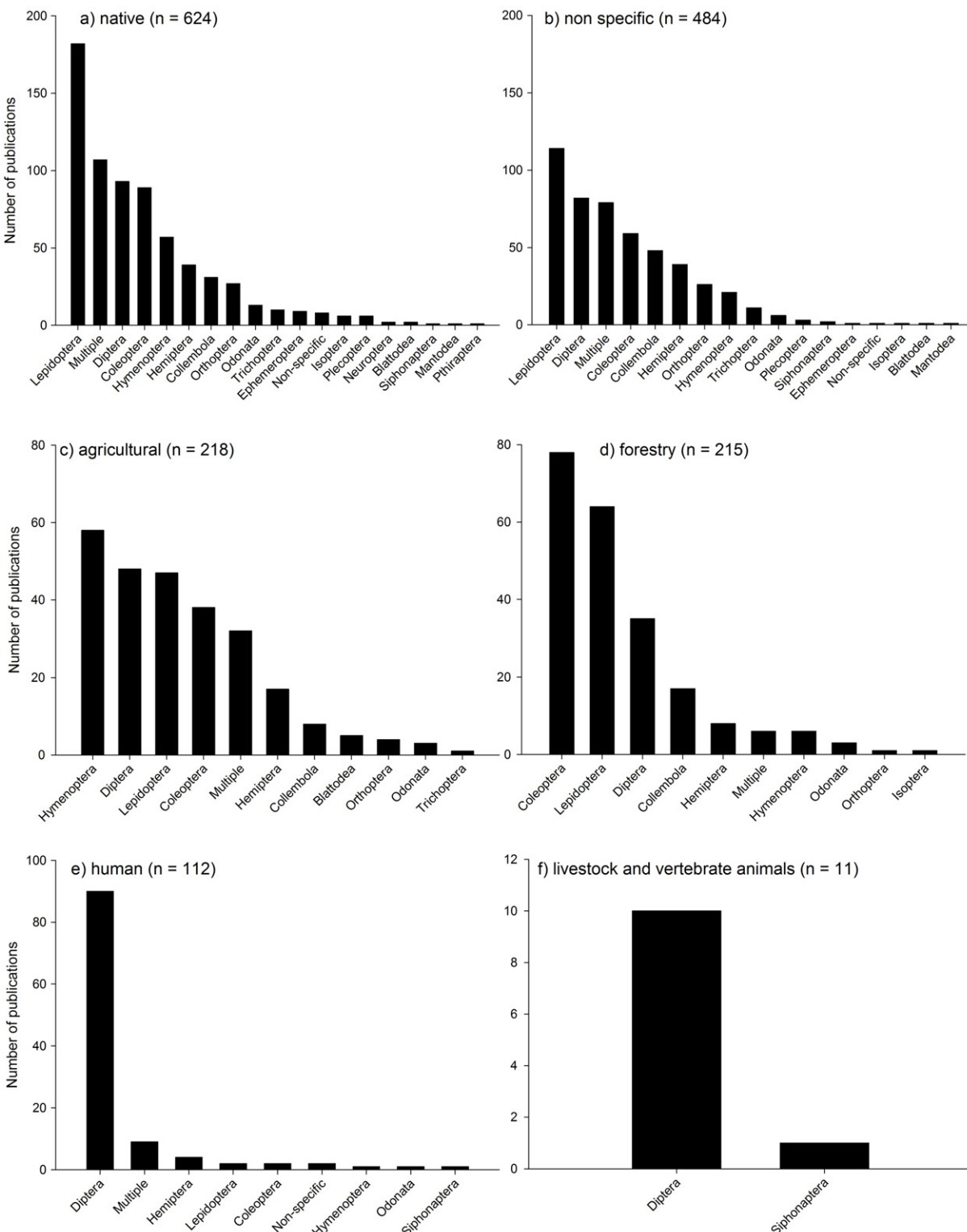

**Figure 6** Number of publications produced for each of the dominant Orders in different habitats. Habitats based on Table 1.

and vertebrate animals (Fig. 6f). In both native habitats and habitats not defined ("non specific"), Lepidoptera were the most dominant taxa assessed, followed by a range of taxa including Diptera, Coleoptera, and multiple taxa. In agricultural systems, Hymenoptera, Diptera, Lepidoptera, Coleoptera and multiple taxa dominated most studies. In Forestry habitats, Coleoptera dominated, followed by Lepidoptera, Diptera and Collembola. In studies assessing human interactions with insects, Diptera dominated, as they did with studies on livestock and vertebrate animals. It is not surprising that there is a relative bias towards some groups such as Lepidoptera (mainly butterflies), and Diptera (primarily *Drosophila*) due to the primarily role that they play in scientific research worldwide, the relative ease of identification (butterflies) and the role that non-drosophilid Diptera play as disease vectors.

We found the five most dominant response variables to climate change were changes in abundance, distribution/range shift, interactions, assemblage composition, and phenology (Fig. 7a). For all response variables (except "non-specific"), data collection was most common, followed by modelling studies, reviews, desktop studies and no theme studies. This trend did not continue for studies on interactions, no theme ("non-specific"), and behaviour where review studies were more common than modelling studies (Figs. 7b and 7c). When response variables were assessed in relation to the 10 top ranked Orders (Figs. 8a and 8b) studies comparing most of the response variables were most studied in the Lepidoptera, Diptera, Multiple taxa, and Coleoptera.

This analysis of current climate change research, specifically on insect groups and the type of information gathered indicates that the Orders Lepidoptera, Diptera, Orthoptera and Collembola are relatively well studied, whilst Coleoptera and Hymenoptera are relatively understudied relative to the number of species currently identified in each Order. There is a general tendency for research publications to focus on population changes in abundance and distribution/range shift, and relatively fewer focusing on biological reasons for these changes. Fundamental biological research areas such as physiology, behaviour, and genetics, are relatively depauperate in the assessment of insect responses to climate change (14% of publications together in total), and this gap could be limiting the ability for broad generalisations to be developed quickly (*Chown et al., 2010*). The importance of insect physiology and behaviour in the distribution and abundance of species has been known for a long time (*Andrewartha & Birch, 1954*), as has the strong association between ecology and physiology (*Chown, Gaston & Robinson, 2004*; *Gaston et al., 2009*). Understanding responses to climate change is a crucial area where knowledge across disciplines is vital (*Williams et al., 2008*). This information is required to identify how scientists can build mechanistic models of responses to climate change (*Angilletta & Sears, 2011*), as well as integrate this information into broader models crossing trophic levels (*Mitchell & Angilletta, 2009*). Furthermore, the integration of models into data collection research studies would also assist in assessing insect responses to climate change (*Angilletta & Sears, 2011*).

Climate change studies with insects have primarily focused on changes in abundance, and change in the distribution of species, as these are the easiest measures of change to

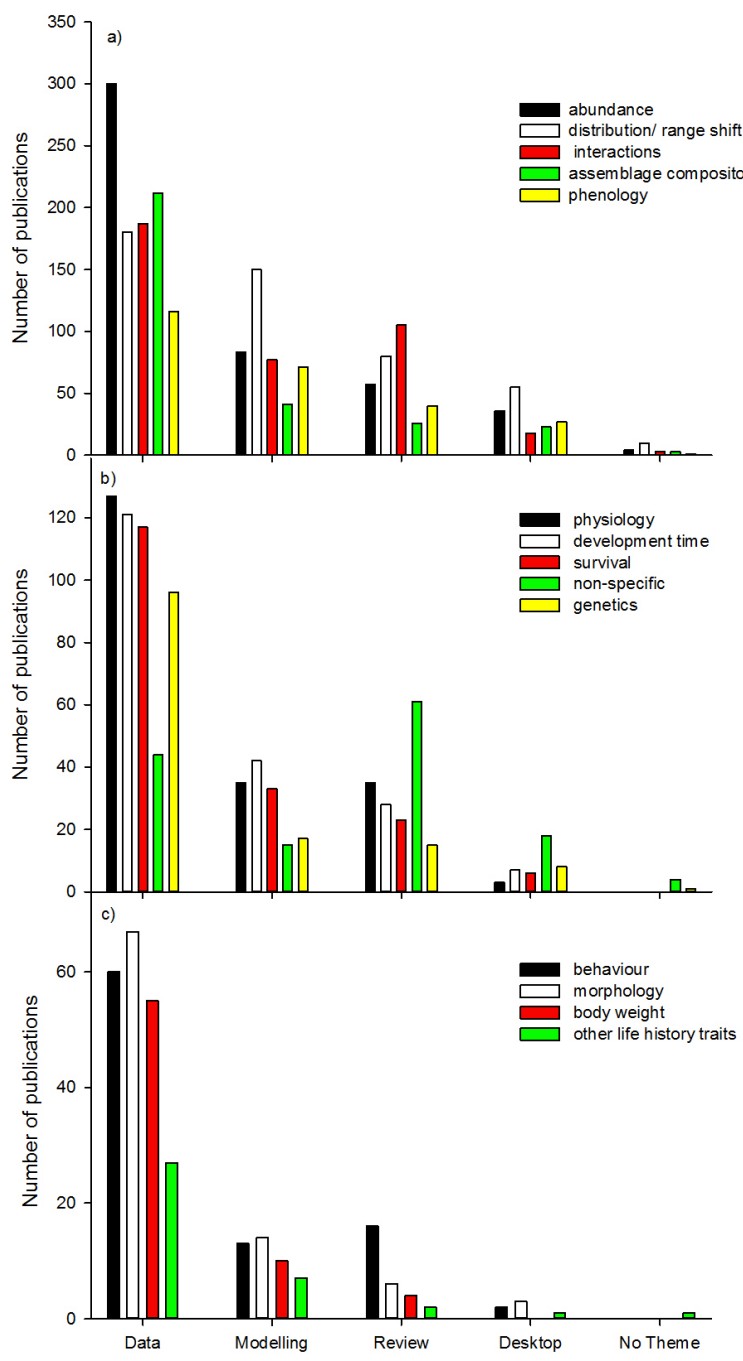

**Figure 7** Publication type and insect response variable recorded in publications. Groupings based on Table 1.

test, especially in field conditions. However, populations may have annual fluctuations and keep within their distribution, but be impacted in other ways including a reduction in fitness and population viability (*Lane et al., 2012*). Surviving insects may be of a particular genotype, or have been pushed closer to their physiological tolerance. An analogous impact of this can be seen in agricultural production areas where prophylactic sprays

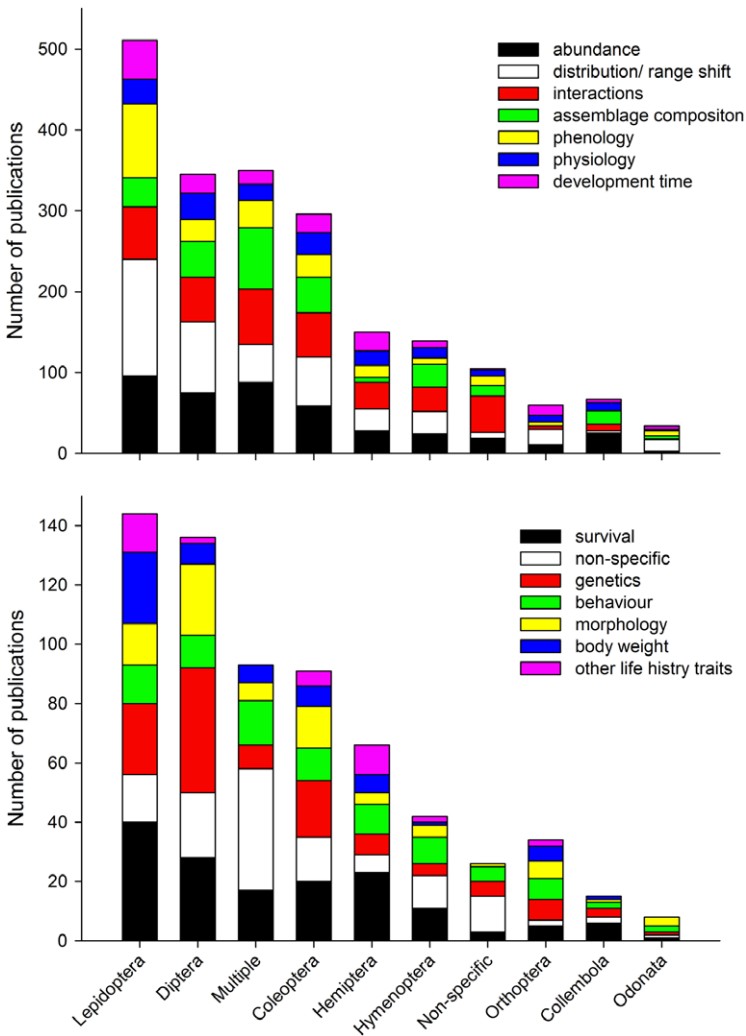

**Figure 8** Number of publications based on the top 10 ranked Orders (based on number of publications) and response variable of taxa studied by authors. Response variables based on Table 1.

have been used to control insects, killing off the majority of individuals, but a small number of a resistant individuals survive (*Gray, Ratcliffe & Rice, 2009*). The resistant animals then increase in numbers and return to high abundances over a relative short time (a few seasons in some cases) with higher resistance to the chemical spray, but with a much reduced genotypic variation (*Page & Horne, 2012*). From a climate change perspective, more thermally tolerant populations within a species may be more resistant than others (*Elmes et al., 1999*; *Nielsen, Elmes & Kipyatkov, 1999*; *Mueller et al., 2011*). Such population changes exemplifies the critical role that genetic and genomic assessments of population responses to climate change play (e.g. *Zakharov & Hellmann, 2008*; *Telonis-Scott et al., 2012*).

Physiological tolerances of thermal extremes vary for different species and for different species across their range. Indeed even within a population, males and females may exhibit

different responses to thermal stress. A non-insect example of this is the pseudoscorpion *Cordylochernes scorpioides* (*Zeh et al., 2012*) put under a 3.5 °C temperature stress (compared to average temperature) using a split brood experiment; the higher temperature reduced development time but also reduced size, particularly of males and thus reducing sexual dimorphism. The males produced 45% less sperm and females failed to produce embryos. However, when males were re-acclimated back to the ambient temperature seven days before mating, sperm count did not change. However, females which did not produce embryos at the higher temperature were able to produce embryos after a seven day acclimation at control temperatures, and no females that were moved from the ambient temperature to the higher temperature became gravid with seven days exposure. Such cases exhibit the crucial importance of understanding fundamental biology of organisms if we are to make accurate predictions about population responses to a rapidly changing climate.

Ectotherms may also stay within their habitat when there are a variety of microhabitats to choose from. Most species modelling is based on average temperature readings that are not directly in the habitat of the organism being studied. For example, WORLDCLIM derived climate data (*Hijmans et al., 2005*) are from weather stations located a few metres off the ground, in open, clear microhabitats not directly associated with the primary habitat of organisms. Therefore the use of such data may lead to very different interpretation than from data collected from an organism's habitat. For example, the warming tolerance (WT) and thermal safety margins (TSM) of the meat ant *Iridomyrmex purpueus* (Andrew et al. unpublished data) based on weather station annual average temperatures were 25.8 °C (WT) and 13.78 °C (TSM). These values declined to 19.5 °C and 7.51 °C when based on summer average temperatures, and declined further to 7.81 °C (WT) and −4.2 °C (TSM) when based on temperatures recorded at the nest site during summer between 10 am and 4 pm when the ants are under the highest thermal stress.

There is a fundamental difficulty in interpreting biological responses to rapid climate change when the biology of species is not well known and experiments are only carried out over short time periods. Most studies which are considered long term usually have no more than 10 years of accumulated data, with older morphological and potential DNA data for single species found in museum collections (*Lister, 2011*). Very few directly comparable collected datasets have data constantly recorded over multiple decades and centuries, keeping consistent methodologies, collecting sufficient data, or sampling comprehensively across all biota (*Magurran et al., 2010*). Two of the most comprehensive datasets we currently have for insects are from Rothamstead in the UK, and of locust outbreaks in China. In Rothamstead, moth records date back to 1933 with light traps at 80 sites around the UK, and aphid populations have been monitored via suction traps since 1964 when a national network of sixteen aphid collection sites were dispersed throughout the UK (*Harrington & Woiwod, 2007*). In China, migratory locusts records were haphazardly collected over a 1,910 year period (*Tian et al., 2011*) from historical documents. The general lack of long-term datasets globally and across taxa challenges our ability to make strong comparative generalisations about species responses and the biological impacts of climate change.

It is clear from the current assessment that changes in abundance and range/distribution shift of single species are the main focus in insect responses to climate change. Therefore for future studies, it is vital that researchers identify the type of habitat they are working in to continue this type of work ('non-specified' habitats accounted for 28% of publications in this assessment). We also recommend an emphasis on the causes of these changes is required by the assessment of insect population dynamics, ecology, physiology, behaviour, and genetic/genomic analyses. Many species will more likely adapt to a changing climate *in-situ*, and what impact this has on their recruitment and population dynamics is unclear. When single species studies are conducted it would be instructive to explore modeled predictions in research papers to identify the impacts of a changing climate on insects. In its most simplistic form species predictions could be undertaken using species distribution modeling or vulnerability indices (*Rowland, Davison & Graumlich, 2011*). When assemblages are assessed then community change predictions would also be of use. This can be done using relationship changes between trophic levels of species and competition among species using existing tools such as game theory (*Mitchell & Angilletta, 2009*), or using statistical relationships based on changes in assemblage composition (fitting predictive models in mvabund *Wang et al., 2012*) or functional traits (Brown AM, Warton DI, Andrew NR, Binns M, Cassis G & Gibb H, unpublished data).

Even today we are still just beginning to understand how insects will respond to a rapidly changing climate, but the current trends of publications give a good basis for how we are attempting to assess insect responses. There is a crucial need for a broader study of ecological, behavioural, physiological and life history responses to be addressed across a greater range of geographic locations, particularly Asia, Australia/Oceania, Africa, and South America, and in areas of high human population growth and habitat modification.

We are not advocating that research should be done on under-represented taxa to merely equilibrate the research done on each taxa relative to the number of identified species. A wider range of taxa should be studied to attempt to identify how generalised trends in response to climate change are across taxa. This also exemplifies the need for understanding the role of phylogenetic relatedness and functional/morphological traits across a range of species and within/among communities (*Yates & Andrew, 2011*; *Srivastava et al., 2012*; *Best, Caulk & Stachowicz, 2013*). Future work should consider these issues critically, and some exemplar questions to pose include: Are the range movements measured in European butterfly's in native habitats, an umbrella response to other more-poorly understood taxa from Europe (including Coleoptera, and Hymenoptera) from those same habitats? Is this trend applicable in other parts of the world? Will individual species, taxa or functional groups respond in a highly variable way within and among regions? Rigorous testing of such predictions are required globally to develop a better understanding of biotic responses to climate change.

## ACKNOWLEDGEMENTS

Graham Hall commented on an earlier version of the manuscript.

### Funding

This research was funded in part by Australian Research Council Discovery Grants (Australia) DP0769961 and DP0985886 to NRA and an Australian Endeavour Research Fellowship to M-PJ. The funders had no role in study design, data collection and analysis, decision to publish, or preparation of the manuscript.

### Grant Disclosures

The following grant information was disclosed by the authors:
Australian Research Council Discovery Grants (Australia): DP0769961, DP0985886.

### Competing Interests

Nigel R. Andrew is an Academic Editor for PeerJ. There are no other competing interests.

### Author Contributions

- Nigel R. Andrew conceived and designed the experiments, performed the experiments, analyzed the data, wrote the paper.
- Sarah J. Hill performed the experiments, wrote the paper.
- Matthew Binns performed the experiments and analyzed the data.
- Md Habibullah Bahar, Emma V. Ridley, Myung-Pyo Jung, Chris Fyfe, Michelle Yates and Mohammad Khusro performed the experiments.

### Supplemental Information

Supplemental information for this article can be found online at http://dx.doi.org/10.7717/peerj.11 and http://dx.doi.org/10.6084/m9.figshare.105599.

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
