# Peer review of "Assessing insect responses to climate change: What are we testing for? Where should we be heading?"

_PeerJ, doi:10.7717/peerj.11_

## Round 0.1 · original submission · Major Revisions

Both reviews suggest revisions, and I concur with the reviewers. In particular I concur with the review suggesting major revisions as the reviewer was very detailed in his assessment.

That reviewer states, among other important and valid points, that "(the) major criticism with the Methods is that readers have no access to the actual data set as a supplemental file or online archive." This is an important point in this era of open data. I encourage the authors to deposit the details of the search in an online repository such as FigShare. Normally I would ask for an augmented reference section in this paper, but that would make the paper very long. Of course, since this is an online publication length is of no importance. So the authors could consider depositing the information within the paper in the literature cited section. In addition, the M&M section needs more overall information on which papers were used (as mentioned by the reviewer). This can easily be accomplished by deposition of the entire bibliography, or annotated bibliography, in FigShare. I would suggest (and this is just a suggestion) to deposit a spreadsheet version of the full list of literature used in this study with column headings such as date, authors, journal, title, species, keywords, and a short description of relevant details. That type of deposited data would be useful for others performing similar meta-analyses and/or those who would like to expand upon this work immediately or in the future.

The reviewer who suggests major changes notes that"

"t is not surprising that there have been proportionally more studies on Lepidoptera and Diptera. Butterflies are one of the very few insect groups for which species-level identification is feasible, whose ranges are well-known and easily tracked, and for which there is reliable long-term data. The focus on Diptera also makes sense because their disproportionate role as disease vectors."

This is true. I would also add that (at least anecdotally, in my experience – although those two make up >25% of the studies recorded in this meta-analysis) much of the climate-change and/or ecophysiological work in the same vein tends to focus either on insects of economic and/or environmental concern in forestry, agriculture, or medicine – or on model organisms. This tendency is also bound to skew the data to a great extent.

The authors state that:

"Fundamental biological research areas such as physiology, behaviour, and genetics, are relatively depauperate in the assessment of insect responses to climate change..."

This is moderately true, although (again anecdotally) the papers of that sort that do exist seem to "punch above their weight." That is, many of those papers are quite influential in the field, though they may not be in the majority within the authors' categorization. I'd also add that this tendency is changing with the advent of easily available genomic data for more and more species. The mountain pine beetle is but one example of this. And the spruce budworm is following close in its stead. Both have had, or are having, substantial genomic data developed. And concurrent with that, both have, or will, see a substantial increase in papers of this sort. This is particularly the case since the range of both extends into areas that are or will be affected by climate change. In the case of mountain pine beetle, we are already seeing a range expansion that is, arguably, due at least in part to shifting climate variables.

These are two species that I am particularly well aware of due to my personal research interests. In this era of burgeoning genomic data, I doubt that they are the only two.

This paper does represent an important part of the discussion on the issue of entomological research and insect conservation. For instance, this work points out areas that are potentially ripe for further investigation. It also pertains well to pest and disease management, as many insect pests and disease vectors will (or are!) shift(ing) their distribution due to climate change (e.g. mountain pine beetle). So, I encourage the authors to carefully consider the reviewers' comments, particularly the reviewer who suggested major revisions, and revise their paper accordingly. In particular, please note the various philosophical points suggested by that reviewer, as his expertise in this area is extensive.

·

Basic reporting

The manuscript is well written and appears to meet basic reporting requirements with a couple of minor omissions: there is no "Conclusions" section, though that may not be necessary as conclusions are well handled in the "Results and Discussion" section. The "Acknowledgments" section is not meant to acknowledge funding sources: I believe that is meant to be handled in a separate Funding Statement on the published paper.

A possible typo appears in line #124 (page 6: "...native habitat assessing temperature..." should probably read: "...native habitat assessed temperature..."

Experimental design

No comments

Validity of the findings

Interesting findings and a valid contribution to the field.

Additional comments

This is a good piece of research and a valid contribution to the field of insects and climate change. With the exception of a couple of very minor modifications (check the language on line #124, remove funding source from Acknowledgements) I believe this manuscript is publication ready.

·

Basic reporting

Many readers would welcome a review on the state of the art in the rapidly expanding literature on climate change impacts on insects. As such, there is valuable information in this study. However, the way in which the data are presented (or not, as the case may be), as well as some unsupported conclusions and recommendations seriously weaken the manuscript in its current form. Several instances of awkward working and grammatical errors detract from the manuscript. Careful proofreading and editing is needed.

The analysis is restricted to the generation of bar graphs. While this is, in itself, may be adequate to get the message across, the large number of graphs included in the manuscript make the exercise seem very exploratory. The number of graphs could be cut by at least 50% and the message would still be clear: there has been more effort to date on selected subsets of orders and regions, and the studies span a range of responses, ecosystem functions and trophic relationships. Also on the figures: Figure 2 and Figure 7 as submitted are exactly the same graphs.

The results section is mostly a readout of the values on the graphs, and could be cut down considerably. The text could focus on overall trends, rather than restatement of values that are readable from the figures.

The references cited in the text are a mix of important papers (e.g. Parmesan, some recent Nature papers), and very focused, context-specific studies. However, several major recent papers are omitted. A few examples: key papers on butterflies by Kerr and colleagues and some very recent papers on changes in bee communities. I assume these are in the larger database of papers considered, but we have no way of knowing that (see below).

Experimental design

My major criticism with the Methods is that readers have no access to the actual data set as a supplemental file or online archive. The methods used in selecting papers are described, but we have no indication as to which papers were used. Without the data there is no way to assess the results. Beyond this major criticism, the Methods section is not clearly written. It would benefit from careful proofreading and rephrasing in some particularly confusing passages (e.g., lines 77-84).

Validity of the findings

It is not surprising that there have been proportionally more studies on Lepidoptera and Diptera. Butterflies are one of the very few insect groups for which species-level identification is feasible, whose ranges are well-known and easily tracked, and for which there is reliable long-term data. The focus on Diptera also makes sense because their disproportionate role as disease vectors. To me, it makes perfect sense that they are “overrepresented”. Similarly the Coleoptera and Hymenoptera are likely underrepresented because the bulk of their global diversity is in the tropics, often in cryptic habitats, and where there have been very few studies of climate change impacts overall. I see no compelling reason why the distribution of studies should be expected to mirror the number of described species in an insect order, and the authors do not provide convincing arguments in support of this.

It is also not terribly surprising that there have been fewer studies of climate change impacts in human-dominated ecosystems. Even if change is documented in such systems, it will be more difficult to disentangle the effects of climate change from other drivers. The picture may simply be “cleaner” in less altered habitats.

The recommendations for future focus are, as the authors acknowledge, speculative. Unfortunately, they are also not convincingly argued, nor are the example taxa necessarily the best match to the stated goals. Furthermore, some of the taxa and systems proposed for further work are precisely those that already make up a disproportionate percentage of published studies. For example, most insects implicated in human and animal health are Diptera, many forest pests are Lepidoptera, and both these orders are often dominant pollinators, especially in arctic and alpine systems where the effects of climate change will likely be most pronounced. The recommendations themselves may be distilled down to “do more research on things we haven’t done much research on”. Overall, the recommendations for future work seem primarily focused on leveling the playing field – smoothing out the bars in the various graphs. But why is it necessary for us to be so focused on doing so? Given the scope and scale of both insect diversity and climate change impacts, and the relatively low numbers of specialists and limited funding, should we not be focusing our efforts on the taxa and systems where the potential rewards in terms of rigorous data are greatest? If we are to draw out a road map for future research programs, it should be as clear, comprehensive and well-justified as possible.

---

## Round 0.2 · accepted · Accept

Thank you for your responses to the reviewers and to my comments as well. I believe that you have adequately made your case where necessary, and have made meaningful changes that make this an even better manuscript.

I do note a number of small typos and grammatical errors, some of which seem to have been incorporated accidentally in the revisions. Please work with the production staff to ensure the high quality of the final published paper.